# Deep Eutectic Solvent-Based Ultrasound-Assisted Strategy for Simultaneous Extraction of Five Macamides from *Lepidium meyenii* Walp and In Vitro Bioactivities

**DOI:** 10.3390/foods12020248

**Published:** 2023-01-05

**Authors:** Keke Li, Zhongyu Li, Lei Men, Jiwen Li, Xiaojie Gong

**Affiliations:** 1College of Life Sciences, Key Laboratory of Biotechnology and Bioresources Utilization, Dalian Minzu University, Ministry of Education, Dalian 116600, China; 2School of Biological Engineering, Dalian Polytechnic University, Dalian 116034, China

**Keywords:** deep eutectic solvents, macamides, *Lepidium meyenii* Walp., ultrasound-assisted extraction, macroporous resin, anti-inflammatory, neuroprotective

## Abstract

This study aimed to develop an integrated approach of deep eutectic solvent-based ultrasound-assisted extraction (DES–UAE) to simultaneously extract five major bioactive macamides from the roots of *Lepidium meyenii* Walp. Ten different DESs containing choline chloride and selected hydrogen-bond donors were prepared and evaluated based on the extracted macamide content determination using high-performance liquid chromatography (HPLC). Choline chloride/1,6-hexanediol in a 1:2 molar ratio with 20% water exhibited the most promising extraction efficiencies under the optimized parameters verified using single-factor optimization as well as Box–Behnken design. Using the optimized DES–UAE method, the extraction efficiencies of the five macamides were up to 40.3% higher compared to those using the most favorable organic solvent petroleum ether and were also superior to those of the other extraction methods, such as heating and combination of heating and stirring. Furthermore, using the macroporous resin HPD-100, the recoveries of the five target macamides from the DES extraction reached 85.62–92.25%. The 20 μg/mL group of the five macamide extracts showed superior neuroprotective activity against PC12 cell injury than that of the positive drug nimodipine. The macamide extracts also showed higher NO inhibition in LPS-stimulated RAW264.7 cells. Thus, the developed approach was a green and potential alternative that can be used to extract bioactive macamide constituents from *L. meyenii* in the pharmaceutical and food industries.

## 1. Introduction

*Lepidium meyenii* Walp., often referred to as “maca”, is an affiliate of the Brassicaceae family and is predominantly grown in the Peruvian Central Andes [1] and the Qinghai–Tibet Plateau in China [2]. In a traditional sense, this plant has been listed as a raw ingredient for medicinal products and foodstuffs [3]. Considering the introduction of maca into the food–medicine continuum as an economic crop, it has been used for various treatments, such as improving fertility and sexual function, promoting gastrointestinal peristalsis, increasing glucose uptake, and promoting antimicrobial activity, as well as for their hepatoprotective, antioxidative, anticancer, neuroprotective, wound healing, antifatigue, and pesticidal effects [4]. Maca root powder has been formally approved to be included in the New Resource Food Directory by the Chinese Ministry of Health in 2011.

The chemical composition of maca roots is complex, and various compounds with biological activity have been found. Phytochemical studies of maca roots have revealed that their predominant bioactive constituents include alkaloids, polysaccharides, glucosinolates, sterols, and essential fatty acids [3]. Among these active compounds, it is widely accepted that maca alkaloids exhibit neuroprotective characteristics [5,6] and exert anti-inflammatory, anti-depressant, and analgesic effects [7]. Macamides, as a type of maca alkaloids, were considered an active ingredient and played a fundamental function in the efficacy of maca for the treatment of neurological disorders and inflammatory disease [8]. As reported in our previous study [9], N-(3-methoxybenzyl)-linolenicamide (C1), N-benzyl-(9Z,12Z,15Z)-octadecatrienamide (C2), N-(3-methoxybenzyl)-(9Z,12Z)-octadecadienamide (C3), N-benzyl-(9Z,12Z)-octadecadienamide (C4), and N-benzyl-hexadecanamide (C5) are the five major macamides present in maca roots. A series of studies demonstrated that macamide compounds have a diverse spectrum of bioactivities, such as antioxidant, antitumor, and antifatigue functions [10,11,12]. Researchers had reported that the extract of maca, which contained abundant macamides including C2, C4, and C5, could be used as a food additive for humans on some reproductive traits in men and women [13,14,15], as well as on Japanese quail for reproductive traits [16].

Currently, achieving a high-quality extraction of the aforementioned macamides from maca roots is crucial in exploring their activities and prospects for usage. Macamides are the secondary metabolites of N-benzylamides alkaloids with a low-polar benzyl group and long-chain fatty acids, and traditional extraction methods generally involve solvent reflux extraction and ultrasound-assisted extraction (UAE) by organic solvents, such as *n*-hexane [17], petroleum ether [9], 80% aqueous acetone [18], and methanol [19]. However, the extraction efficacy is limited, and most of the organic solvents show intrinsic disadvantages including high toxicity, strong volatility, and non-degradability, which are harmful for human health and the environment [20]. Recently, researchers have been interested in the development of green chemistry and technology as an ecofriendly process [21]. Abbot et al. [22] developed the concept of deep eutectic solvents (DESs), which are made up of hydrogen-bond donors (HBD) and hydrogen-bond acceptors (HBA). Owing to DESs being easy to prepare, inexpensive, biodegradable, and environmentally friendly, they have become ideal alternatives to conventional volatile organic solvents and are identified as potentially green and sustainable solvents for scientific and industrial use [23,24]. Several reports have described the application of DESs in extracting and separating multiple natural products from a variety of polar and non-polar natures [25,26,27,28]. To the best of our knowledge, no research has been conducted on the simultaneous extraction of the aforementioned major macamides using DESs in addition to typical organic solvents.

UAE mainly depends on mechanical effects, such as strong oscillation and cavitation, to effectively infiltrate a solvent into the plant cell wall, which can significantly improve extraction efficiency [29]. UAE has attracted significant attention as an extraction technique owing to its high efficiency, rapidity, and low solvent consumption for different bioactive products [30,31], and is usually coupled with DESs to enhance the quality and quantity of the extracted materials [32,33,34]. The DES–UAE approach, a revolutionary technology, shows considerable promise for applications in the pharmaceutical, cosmetic, and food industries.

Thus, the aim of this study was to obtain the bioactive extracts of the five macamides from maca roots using the DES–UAE technique. For evaluating extraction efficiency, several choline chloride-based DESs were produced and subjected to strict testing. The process parameters of UAE were optimized via the response surface methodology (RSM). Box–Behnken design (BBD) based on the single-factor optimization validated the optimal parameters. The target macamides from maca roots were analyzed using high-performance liquid chromatography (HPLC). The recovery of the five major macamides from the DES extracts was evaluated using a series of macroporous resins. Furthermore, this novel approach was compared with established procedures. The neuroprotective activity against PC12 cell injury and the anti-inflammatory activity on RAW264.7 cells were selected to evaluate the bioactivities of the macamide extracts. As an efficient and new preparation method for macamide extracts, its performance in biological activity determines its application value. It is worth studying the bioactivities of our newly prepared macamide extracts, although the activities of some of the monomeric macamides they contained have already been approved [4]. This is the first developed technique for extracting low-polar macamides from maca roots; moreover, the developed technique has practical utility in the pharmaceutical, nutrition, and food industries.

## 2. Materials and Methods

### 2.1. Materials and Chemicals

Fresh L. meyenii (maca) samples were collected from Shannan city, Tibet, China, in December 2016. The samples were identified by an author of this study (Xiao-Jie Gong) and a voucher specimen (No. 2018005) was deposited at our laboratory. After removing the sediment and washing with water, they were vacuum dried for 24 h at 40 °C in an oven (DZF–6030, Jingmi instrument Co., Ltd., Shanghai, China). Then, the maca roots were powdered using a laboratory grinder (JP–1000C–8; Yongkang instrument Co., Ltd. Yongkang, China), passed through a 40-mesh sieve, and stored in fastened glass bottles at 4 °C for additional analysis. Macroporous resins (DM-130, NKA, HPD-100, D-101, HP-20, D4020, AB-8, and DM-301) were obtained from Haoju resin technology Co., Ltd. (Tianjin, China). Unless otherwise stated, all reagents used were of analytical grade.

### 2.2. DES Preparation

The DESs were synthesized via heating several constituent blends to 80 °C and constant stirring to form a homogeneous liquid [26], and when required, a known amount of water was added. Table 1 lists the HBA, the HBD, the molar ratios of HBA/HBD, and the abbreviations of the DESs prepared in this study.

### 2.3. HPLC Quantification

For HPLC analyses, the Agilent 1260 HPLC system was fitted with an ultraviolet–visible detector (DAD), a quaternary pump, and an online vacuum degasser. Chromatographic separation was performed using a Zorbax XDB C18 column (5 µm, 4.6 mm × 250 mm). The compounds were eluted for 30 min in water (solvent A) and acetonitrile (solvent B) with a 20:80 (A:B) to 100 (B) gradient. Then, 10 µL of the analyte was injected at a flow rate of 0.8 mL/min and column temperature of 40 °C. Subsequently, the extract was filtered through a membrane filter (0.45 μm; Whatman, NJ, USA), followed by injection. The wavelength for detection was 210 nm [9,15]. The actual HPLC chromatograms of the standard references and the DES extracts are shown in Appendix A.

The calibration curves were generated to analyze the contents by plotting the peak area vs. concentration using six data points. The limits of detection (LOD) and the limits of quantification (LOQ) were identified with S/N (signal-to-noise) ratios of 3 and 10, accordingly.

The reproducibility and accuracy of the underlined assay procedure were confirmed through both intra- and inter-day variability. In case of intra-day precision, a sample solution along with a standard solution was evaluated six times on the same day. The inter-day precision was evaluated by analyzing the sample and standard solutions (as in intra-day) on three different days, followed by calculating the % RSD values for both intra- and inter-day precisions of the five target macamides.

The test for stability was conducted by evaluating one extracted sample throughout 12 h up to 2 days. The % RSD was then calculated for the evaluation of precision and reproducibility.

### 2.4. Extraction Procedure

For the DES-based UAE of macamides (C1–C5), 1 g of powdered maca roots was blended with 10 mL of DESs in a 20 mL centrifuge tube. Then, DES-based UAE was performed in an ultrasonic bath at an ultrasound voltage level of 300 W and a frequency of 40 kHz. The temperature was controlled within ±0.5 °C with a calibrated thermometer and adjusted with ice water. After extraction for 30 min at 40 °C, the supernatant was collected for HPLC analysis by centrifuging the obtained solution for 10 min at a centrifugation rate of 17,000× *g*. For comparison, UAE using conventional organic solvents was also carried out under the same parameter as that of the DES-based UAE.

Based on the results of the DES-based UAE, the optimal extraction efficacy of the DESs was selected for further solvent screening. The effect of the molar ratio of HBA and HBD (1:1, 1:2, 1:3, 1:4, and 1:5) and water content (0, 10%, 20%, 30%, and 40%) on the extraction efficacy of selected DES were evaluated. The sonication input power and centrifugation procedures were the same as described above. Each extraction procedure was performed in triplicate.

### 2.5. DES–UAE Process Optimization

#### 2.5.1. Single-Factor Optimization

Single factor optimization was used for optimizing the DES system and extraction settings. In this experiment, 10 different kinds of DESs were employed to determine the optimal solvent for macamide extraction from maca roots. Then, the molar ratio of choline chloride to 1,6-hexanediol (1:1, 1:2, 1:3, 1:4, and 1:5) and the concentration of water in the DESs (0, 10%, 20%, 30%, and 40%) were optimized to determine the DES system. Moreover, the extraction power (100–300 W), solution-to-solid ratio (6:1–14:1), duration of extraction (10–50 min), and temperature used in the extraction (20–60 °C) were comprehensively investigated. The extraction yields of the five target macamides were analyzed using the samples in a variety of extraction conditions.

#### 2.5.2. BBD Optimization

A three-factor, three-level (−1, 0, +1), BBD combined with RSM was conducted to investigate the optimal independent variables of *X*_1_ (solution-to-solid ratio), *X*_2_ (extraction temperature), and *X*_3_ (extraction duration) to the dependent variables of extraction yields according to the results obtained from the preliminary single-factor investigations. Table 2 lists the experimental parameters and the extraction yields obtained from 17 different experimental trials. Using the Design-Expert software (version: 8.0; Stat-Ease, Inc., Minneapolis, MN, USA), this study successfully generated the BBDs and evaluated the experimental outcomes. Multiple regressions were conducted to examine the data obtained from the experimental trials, which were subjected to an additional adjustment to the quadratic polynomial model shown below:(1)$$Y=\beta_{0}+\overset{3}{\underset{i=1}{\sum }}\beta_{i}X_{i}+\overset{3}{\underset{i=1}{\sum }}\beta_{ii}X_{i}^{\;2}+\overset{3}{\underset{i=1}{\sum }}\overset{3}{\underset{j=i+1}{\sum }}\beta_{ij}X_{i}X_{j}$$
where *Y* denotes the dependent variable; *β*_0_ denotes the constant coefficient; and the linear, quadratic, and interaction effects of the variables are represented by the model coefficients *β_i_*, *β_ii_*, and *β_ij_*, respectively. The coded independent variables are denoted by *X_i_* and *X_j_*. The analysis of variance (ANOVA) was performed to determine the statistical significance of the equation according to the *p*-values.

### 2.6. Other Extraction Methods for Comparison

Two other common extraction methods of using a DES as a solvent and heat reflux extraction or heat reflux extraction along with stirring were performed to compare the efficiency of extraction of the optimized DES–UAE method. Briefly, 1 g of maca root powder was refluxed at 60 °C for 1 h, using 10 mL of the optimal DES system as a solvent. Then, UAE was applied for the extraction of the target macamides from the maca roots using the optimal organic solvent PE as a solvent. A total of 10 mL of PE was placed in a 20 mL centrifuge tube that contained powdered maca roots (1 g), and the extraction process was performed at 40 °C for 30 min in an ultrasonic bath at an ultrasonic power of 300 W. Each extraction procedure was performed in triplicate. After extraction, the mixture was filtered and the supernatant was collected for HPLC analysis.

### 2.7. Recovery of Target Macamides from DES Extracts

From the DES extraction, macamide recovery was performed using eight types of macroporous resins, namely AB-8, DM-130, D-101, NKA, HPD-100, HP-20, D4020, and DM-301. First, 10 mL of the DES extraction solution was introduced into a glass column (1 cm × 10 cm) filled with 3 g of resin. The extraction-containing resin column was allowed to rest for 24 h to achieve adsorption equilibrium before the elution of DESs and polar ingredients with deionized water thrice (50 mL each time), followed by continuous elution in 150 mL of 70% ethanol for the target macamide compounds. A rotary evaporator was employed to remove the eluate ethanol at 35 °C in vacuum conditions. Based on the aforementioned approach, the recoveries of each macamide were determined. Notably, each experiment was performed in triplicate to ensure accuracy.

### 2.8. Evaluation of Neuroprotective Activities

PC12 cells were seeded in 96-well plates at a density of 4 × 104 cells/mL in 100 μL of DMEM per well. The normal group was cultured in DMEM with 5% CO_2_ at 37 °C. The model oxygen-glucose deprivation/reoxygenation (OGD/R) group was incubated in DMEM without sugar with 5% CO_2_ and 94% N_2_ at 37 °C for 2 h; then, the medium was replaced with high-sugar DMEM. The extracted macamide products of PE-UAE, DES–UAE, DES-Heating, and DES-Heating + Stirring with high-sugar DMEM (5, 10, and 20 μg/mL each, respectively) were added into wells with 5% CO_2_ and 95% O_2_, and reperfused in an incubator at 37 °C for 24 h [35]. The positive control was selected as nimodipine (10 μg/mL). The non-radioactive cell counting kit-8 test was employed to determine the viability of the PC12 cells after OGD/R injury. The optical density was measured at 450 nm using a microplate reader.

### 2.9. Evaluation of Anti-Inflammatory Activities

The nitrite concentration was assessed using the Griess method [36] by determining the amount of nitric oxide (NO) product in the cultured RAW 264.7 macrophage supernatants. Seeding of RAW 264.7 macrophages was performed in 100 μL aliquots (106 cells/mL) in 96-well culture plates. After overnight incubation, the cells were subjected to lipopolysaccharide (LPS, 1 μg/mL) treatment at different dosages (5, 10, and 20 μg/mL) of the extracted macamide products, followed by further incubation of the plates for 20 h. Then, the supernatants were removed (100 μL) and added to the Griess reagent (0.1 mL; 1% sulfanilamide and 0.1% naphthyl ethylene diaminedihydrochloride in 5% H_3_PO_4_), followed by incubation at ambient temperature for 15 min, and a purple azodye was formed. Using a microplate reader, the absorbance was determined at 570 nm. L-NG-nitroarginine methyl ester L (L-Name, a known NO synthase inhibitor) (2 mM) was used as the positive control.

### 2.10. Statistical Analysis

The data and experimental design were analyzed statistically via the Design-Expert Ver. 8.0 (Stat-Ease Inc., Minneapolis, MN, USA) or GraphPad Prism 8.0, and comparison among groups was performed using one-way ANOVA (analysis of variance) or *t*-test. The Duncan multiple range tests were carried out to determine the differences between the factors in case of significance (*p* < 0.05).

## 3. Results and Discussion

### 3.1. Single-Factor Experimentation

To achieve the optimal yields of the target compounds, the DES solvent system and UAE extraction parameters were optimized using single-factor experimentation.

#### 3.1.1. Screening of DESs

The components of DESs significantly affect their physicochemical characteristics, including polarity, viscosity, and capacity to solubilize, thereby directly influencing the extraction efficiency of target compounds [26,37]. It was observed that the DESs could be successfully formed through heating choline chloride with polyols, whereas, when choline chloride was heated with acids, a stable and transparent liquid could not be formed at a room temperature of 22 °C. Moreover, it was challenging to develop less viscous and clear DESs by heating sugar with acids (e.g., malonic acid, DL-malic acid, fructose, sucrose, and xylose). Aiming for the optimal DESs for the extraction of macamide alkaloids, ten different DESs were selected in this study (Table 1).

As shown in Figure 1A, three types of DESs (DES-1, DES-6, and DES-7) exhibit higher yields of extraction than the others, and only DES-7 exhibits a favorable extraction effect for all five target compounds, particularly for the relatively lower contents of compounds C1 and C2. Among the ten DESs, a favorable extraction performance of DES-7 may be owing to more electrostatic interactions and a superior hydrogen bonding ability with macamides than those of others [15,38]. Thus, DES-7, with a 1:2 molar ratio of choline chloride to 1,6-hexanediol, was selected. Next, five molar ratios of choline chloride/1,6-hexanediol varying from 1:1 to 1:5 were synthesized to explore the effectiveness of the extraction technique. As shown in Figure 1B, an increasing proportion of 1,6-hexanediol in the DESs can decrease the interactions between choline chloride and 1,6-hexanediol, thereby decreasing the extraction yields. The lower or higher ratio of choline chloride to 1,6-hexanediol is not conducive to macamide extraction. As previously reported [26], this phenomenon is likely related to the hydrogen bonding interactions between DESs and macamides compounds and the physicochemical property of DESs. The effect of diffusion and mass transfer improves with increasing the polyol ratio in the DESs, which is owing to a decrease in the surface tension and viscosity, consequently increasing the effect of diffusion and mass transfer. Nevertheless, a further decrease in the ratio of choline chloride would reduce the interactions between the target macamides and chloride anions. Thus, a molar ratio of 1:2 for choline chloride and 1,6-hexanediol was demonstrated to be ideal for the extractability of macamides from maca roots for the next set of experiments.

The addition of water to the DESs led to a decrease in the viscosity of the solvent, and this was favorable for the mass transfer of plant matrix to the solvent [33]. As shown in Figure 1C, DES-7 in water is conducive to the extraction of the slightly higher polar compound, C1, and the extraction yields reach the maximum when water concentration is 20% (*v*/*v*). Nevertheless, a higher water concentration in DES-7 causes a decline in the extracted amounts of the five target macamides. This may be owing to the low-polar properties of macamides, because an increased water content enhances the solvent mixture polarity and decreases the interactions between DES-7 and macamides. Similar results have been reported in the literature [33,39]. At a higher water content, hydrogen bonds are saturated with the abundant amount of water, and insufficient hydrogen bonds are not conducive to extraction. Thus, an increase in water concentration in DES-7 hinders the dissolution degree of macamides. Finally, the optimal concentration of water was observed to be 20% (*v*/*v*). To summarize, the extraction solvent used in this study was DES-7 with 20% of water content.

#### 3.1.2. Screening of UAE Parameters

Ultrasound power was a crucial variable that affected the extraction yields. The extraction efficiency of the five macamides improved significantly when the irradiation power was increased from 100 to 250 W, and subsequently, this increasing trend declined and peaked at 300 W (Appendix A). This is consistent with previous studies conducted on DES–UAE [25]. Therefore, the optimal extraction power was set at 300 W in this study. The remaining parameters, including the solution-to-solid ratio, extraction duration, and extraction temperature, also significantly influenced the extraction yields. As depicted in Appendix A, higher solution-to-solid ratio, extraction temperature, and extraction duration do not facilitate an enhancement in macamide extraction. The extraction procedure was performed at 40 °C for 30 min at a solution-to-solid ratio of 10:1 via the single-factor experimentation.

### 3.2. BBD Optimization of Extraction Process Conditions

A combination of BBD and RSM was conducted for optimizing the DES-based extraction parameters for the macamides, including the solution-to-solid ratio, duration, and extraction temperature. Table 2 lists the results of the experiment obtained from the BBD. Table 3 shows the recorded observations from the ANOVA.

As listed in Table 3, *p*-values across all five models are significantly less than 0.05, demonstrating that the fitness of the model is statistically significant. The lack-of-fit is not significant for the response with *p*-values of 0.1470–0.64644 (>0.05) and the determination coefficient (*R*^2^ ≥ 0.9787) is approximately one for each model, demonstrating that the observed and anticipated yields exhibit a greater level of correlation. Additionally, each adjusted determination coefficient (Adjust *R*^2^) is approximately one, indicating that the experimental values can be accurately predicted by each model. It is observed that the linear terms of *X*_2_ and *X*_3_, the interaction terms of *X*_1_*X*_3_, as well as the quadratic terms of *X*_1_^2^, *X*_2_*^2^*, and *X*_3_^2^, are significant for the responses of all five macamides (*p* < 0.05). The remaining terms are not significant for all five macamides, and all terms are significant for the responses of C1 and C5.

According to the results of the above analyses, the models are expressed as second-order polynomial quadratic equations for the extract yields (*Y*) and coded factors (*X*_1_, *X*_2_, and *X*_3_) as indicated below:Y_C1_ = 79.27 − 1.90*X*_1_ − 1.25*X*_2_ − 0.43*X*_3_ + 1.66*X*_1_*X*_2_ + 1.67*X*_1_*X*_3_ + 1.76*X*_2_X_3_ − 5.94*X*_1_^2^ − 6.47*X*_2_^2^ − 3.86*X*_3_^2^(2)
YC2 = 608.53 − 39.76*X*_1_ − 12.80*X*_2_ − 7.85*X*_3_ + 9.62*X*_1_*X*_2_ + 35.95*X*_1_*X*_3_ + 17.14*X*_2_*X*_3_
− 82.95*X*_1_^2^ − 78.10*X*_2_^2^ − 26.69*X*_3_^2^
(3)
YC3 = 56.67 − 2.76*X*_1_ − 1.31*X*_2_ − 0.91*X*_3_ + 1.48*X*_1_*X*_2_ + 1.71*X*_1_*X*_3_ + 2.22*X*_2_*X*_3_ − 5.90*X*_1_^2^ − 7.06*X*_2_^2^ − 3.20*X*_3_^2^(4)
YC4 = 708.61 − 31.57*X*_1_ − 17.75*X*_2_ − 11.11*X*_3_ + 7.09*X*_1_*X*_2_ + 16.05*X*_1_*X*_3_ + 11.07*X*_2_*X*_3_ − 106.67*X*_1_^2^ − 79.47*X*_2_^2^ − 24.07*X*_3_^2^(5)
YC5 = 1212.89 − 60.63*X*_1_ − 31.23*X*_2_ − 13.16*X*_3_ + 22.22*X*_1_*X*_2_ + 19.25*X*_1_*X*_3_ + 73.02*X*_2_*X*_3_ − 126.45*X*_1_^2^ − 129.17*X*_2_^2^ − 61.55*X*_3_^2^
(6)

Three-dimensional (3D) surface plots were generated for the purpose of visualizing the interactive impacts of the independent factors (solution-to-solid ratio, temperature for extraction, and duration of extraction) on the product yields of the five macamides.

Figure 2a,d,g,j,m show the impact of extraction time and solution-to-solid ratio on the extraction yields of the five macamides. When the extraction time is set to a specific point, the extraction rate of the five macamides increases initially and then decreases with changing solution-to-solid ratio. When the solution-to-solid ratio is constant, the extraction rate of the five macamides is observed to increase slightly and then gradually decrease with increasing time. This indicates that the target compounds are highly appropriate within a specific range of extraction time, and more heat can be absorbed through prolonged extraction time, consequently leading to a decrease in the extraction rate [40]. Therefore, the interactive effect of extraction time and solution-to-solid ratio is observed to be significant.

Figure 2b,e,h,k,n show the extraction yields of the five macamides affected by the extraction time and temperature. Notably, at a specific time, an increasing temperature does not promote the high yields. Owing to the increasing diffusivity, an increase in temperature might reduce the viscosity of DESs, thereby improving the efficiency of the extraction process [41]. However, a continuously increasing temperature decreases the yields, which is likely due to the degradation of macamides at high temperatures [42].

Figure 2c,f,i,l,o show the effects of extraction temperature and solution-to-solid ratio on the extraction yields of the five target macamides; similar patterns to those of extraction temperature and extraction time are observed. When the solution-to-solid ratio is constant, the extraction yields increase consistently and reach a maximum point when the solution-to-solid ratio and extraction temperature are 10:1 and 40 °C, respectively. After this point, the yields decrease as the extraction proceeds. A variation in yield owing to changes in extraction temperature is also observed; at a constant solution-to-solid ratio, high temperatures hinder macamide extraction, which is similar to the observed interactive effect between extraction time and temperature.

### 3.3. Validation of HPLC Analysis

The obtained data from the calibration plots, R^2^, LOQs, LODs, and linear ranges of the five analytes with the established procedure of HPLC are concluded in Appendix A. R^2^ > 0.9992 shows considerable linearity of the five compounds. In the precision study, the obtained results indicate that the intra- and inter-day precisions of the five analytes range from 1.79% to 3.83% and from 1.86% to 4.77%, accordingly (Appendix A). The repeatability results indicate that the RSD is less than 2.05%. The stability test shows that the RSD of the five compounds is less than 4.58% which reveals the sample stability for a couple of two days. The established method demonstrates effective accuracy and reproducibility.

### 3.4. Model Verification

The verification tests were performed in triplicate under the determined optimal conditions to study the suitability of the anticipated response values. Considering the convenience and non-significant differences in these predicted models, the optimal macamide extraction was achieved using DES-7 with 20% of water content under the conditions of 10 mL/g solution-to-solid ratio, 40 °C extraction temperature, and extraction duration of 30 min. The observed and predicted extraction yields of each macamide are listed with a small relative error (δ ≤ 1.54%) in Appendix A.

### 3.5. Comparison of Extraction Efficiency

#### 3.5.1. Petroleum Ether and DESs

Herein, the conventional solvents, including anhydrous ethanol, methanol, ethylacetate, cyclohexane, and petroleum ether (PE), were used to compare to the extraction efficiencies for the five target macamides using DES-7 with 20% water content via the UAE (Appendix A). This study showed the same results as previously reported by us [9], with PE exhibiting the highest extraction yields of macamides among the organic solvents. Notably, the increment of extraction yield of each macamide when comparing the DESs to PE is listed in Table 4. There is an increase of 16.2–56.2% in the extraction yield of each macamide, and the yield cumulation of the five amounts increases by 40.3%.

Existing studies have reported that DES could be used as an ecofriendly alternative powerful solvent with a higher extraction efficiency for the extraction of hydrophobic low-polar products [39,43]. In this study, the optimally prepared DES also showed to be more effective and greener than conventional organic solvent for the extraction of macamides.

#### 3.5.2. Heating, Heating + Stirring, and UAE

In this study, the UAE procedure was compared to the heating and combination of heating and stirring methods using the solvent of DES-7 with 20% water content to extract macamides from maca roots. As depicted in Figure 3, the extraction yields of the target macamides obtained via the UAE are higher in comparison to those of heating and combination of heating and stirring. Using UAE can prevent structural damage due to a long-time process under higher temperatures, which indicates that UAE is an optimal extraction strategy for macamides.

### 3.6. Recovery of Target Macamides

Recovery of a target macamide from DES extraction is complicated owing to the high-water miscibility and low-vapor pressure of DESs. Numerous methods, such as the utilization of C18 SPE, macroporous resins [33,40], and antisolvents [25] have been described for recovering the extracted chemicals via DES extraction. In this study, the simultaneous recovery of the extracted macamide (C1–C5) compounds from the DES solution was carried out based on different types of resin (Table 5) for the first time. HPD-100 is a non-polar macroporous resin and suitable for absorbing the target macamide compounds, and the recovery yields of the five macamides all reach 85% after the elution treatment. The reason is that the polarity of HPD-100 is similar to the five macamides. Hence, HPD-100 is suitable and efficient for the enrichment and recovery of the five target macamides from the DESs.

### 3.7. Pharmacological Activities

Neuroprotection is the most common medicinal application of maca, and the *n*-pentane extract of maca roots and a few specific macamide compounds have been observed to exhibit neuroprotective activity in cells [44,45]. Inflammation is associated with osteoporosis [46], and few macamides possess anti-osteoporotic activities [47,48]. In this study, the neuroprotective and anti-inflammatory activities of the extracted macamide products from different extraction methods, including PE-UAE, DES–UAE, DES-Heating, and DES-Heating + Stirring, were measured for the first time to verify their application potential.

As shown in Table 6, the viability of PC12 cells of the extracted macamide products from the DES–UAE extraction (20 μg/mL) is slightly stronger than that of the positive drug nimodipine (90.67% vs. 87.98%). All the other extracted macamide products show neuroprotective activity compared to the model group, but are poorer than nimodipine. For the neuroprotective activity of four different extracted macamide products at 5, 10, and 20 μg/mL, the DES–UAE extraction is the strongest at all three concentration levels. The above results confirm that the extracted macamide products using the DES–UAE exhibit better in vitro neuroprotective activity on PC12 cells. The extracted macamide products exhibit significant NO inhibition in LPS-stimulated RAW264.7 cells (Table 7). Notably, the DES–UAE extraction shows higher NO inhibition (from 50.46% to 55.73%) than other extracts at all three concentration levels. Our data establish the potential anti-inflammatory activity of the extracted macamide products.

The results in this study demonstrated that the extracted macamide products of the DES–UAE possessed higher neuroprotective and anti-inflammatory activities than those of the PE-UAE, DES-Heating, and DES-Heating + Stirring. According to the contents of the extracted macamide products, it was obvious that the high contents of macamide compounds C1–C5 endowed the superior pharmacological activities of maca roots extracts. As previously reported, the neuroprotection of some macamides has been demonstrated, and the mechanism may involve inhibition of mitochondrial apoptosis, activation of Akt and cAMP-response element binding protein phosphorylation [49], and binding to cannabinoid receptor 1 [45]. The anti-inflammatory effect of macamides is also amazing. Zhu et al. reported that compound C4 could significantly decrease pro-inflammatory factors and reactive oxygen species in mice after a 30 min swimming test [50]. In this study, based on the preparation of the maca extracts using DES–UAE, which were rich in macamides, satisfactory bioactivities were observed. The findings revealed that the extracted macamide products of DES–UAE could be a valuable natural neuroprotective and anti-inflammatory agent and could be developed as functional foods. 

## 4. Conclusions

This study developed a feasible and green DES–UAE methodology for extracting five major macamides from the roots of *L. meyenii*. For all the DESs tested, choline chloride and 1,6-hexanediol in a 1:2 molar ratio containing 20% water content was identified to be the optimal solvent; it was demonstrated to exhibit a higher extraction efficiency compared to the standard organic solvents and numerous other DESs. The optimization of the DES–UAE settings was achieved by performing single-factor experimentation and RSM with an ultrasound power of 300 W, a solution-to-solid ratio of 10:1 mL/g, an extraction temperature of 40 °C, and an extraction duration of 30 min. Under the optimal conditions, five macamides (C1, C2, C3, C4, and C5) extracted using DES–UAE reached the maximum yields of 79.27, 608.53, 56.67, 708.61, and 1212.89 μg/g, respectively. Verification studies conducted under optimal conditions demonstrated that the developed method was applicable and reliable for simultaneously extracting the five macamides from maca roots. Moreover, the macroporous resin HPD-100 helped achieve a satisfactory recovery of the five target macamides from the DES extraction in the 85.62–92.25% range. The extracted macamide products were demonstrated to have potential neuroprotective and anti-inflammatory activities through in vitro bioassays. These results revealed that the DES–UAE technique proposed in this study offers an environmentally friendly, efficient, and promising method for green solvent extraction from maca roots for use in the food and pharmaceutical sectors. Future studies are required to examine the mechanism of DES extraction for macamide compounds and the detailed bioactivities of the macamide products.

## Figures and Tables

**Figure 1 foods-12-00248-f001:**
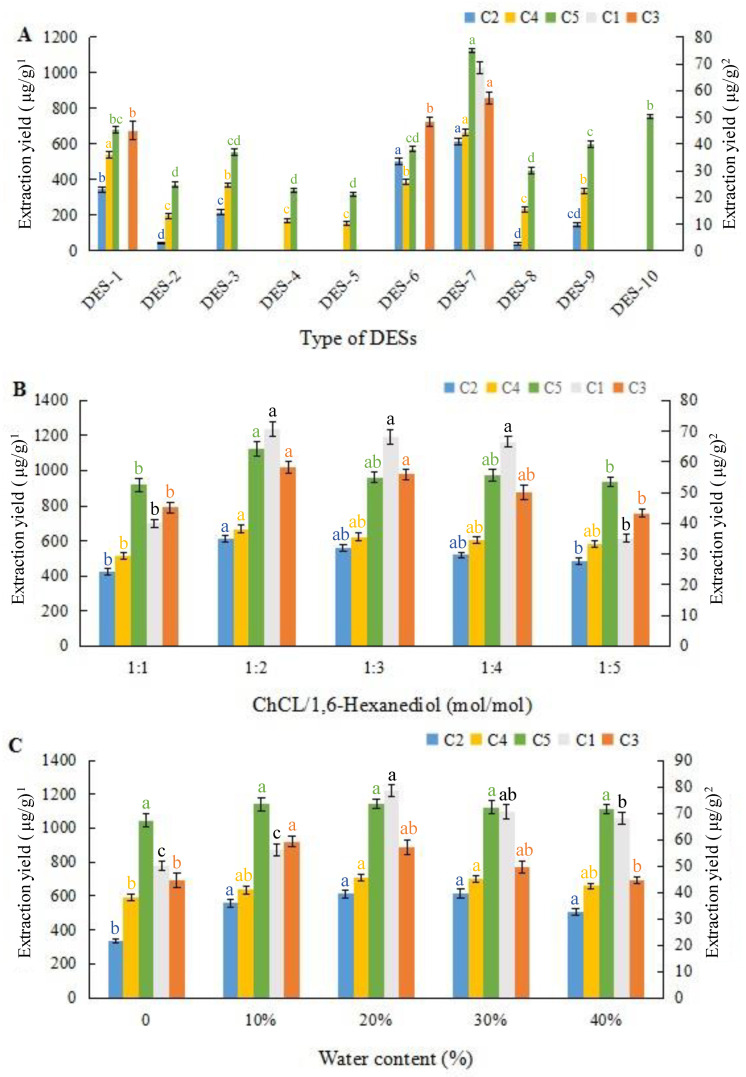
The effects of different DESs (**A**), choline chloride to 1,6-hexanediol ratios (**B**), and water content (**C**) in DESs on the extraction yields of the five macamideds. ^1^ Extraction yields of C2, C4, and C5; ^2^ Extraction yields of C1 and C3. Different lowercase in the columns of the same color represents a significant difference at *p* < 0.05.

**Figure 2 foods-12-00248-f002:**
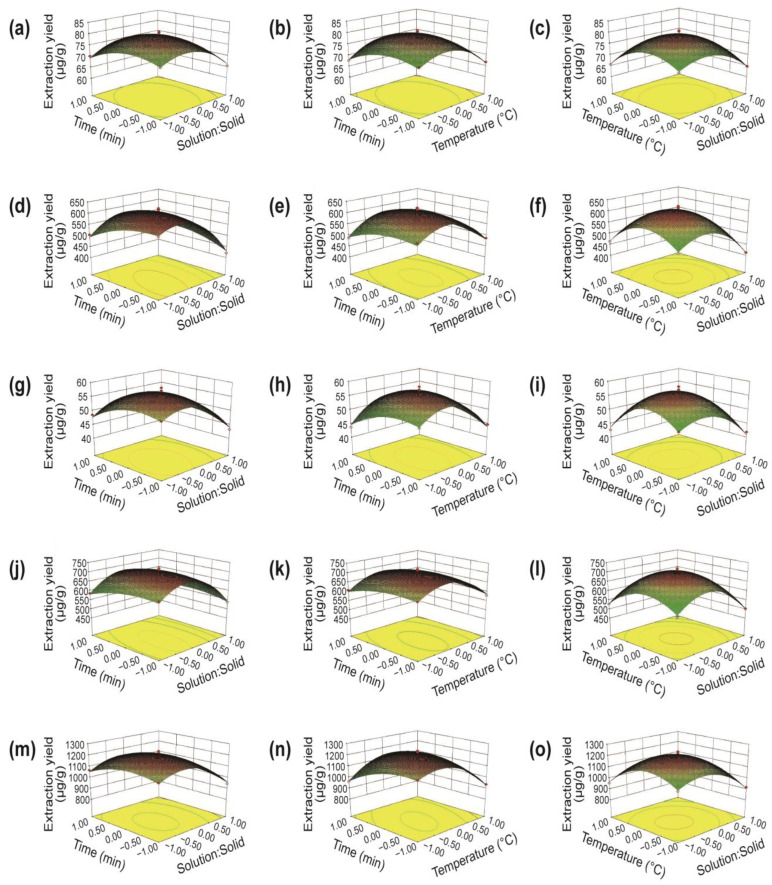
Response surface plots of the models for C1 (**a**–**c**), C2 (**d**–**f**), C3 (**g**–**i**), C4 (**j**–**l**), and C5 (**m**–**o**) of maca roots.

**Figure 3 foods-12-00248-f003:**
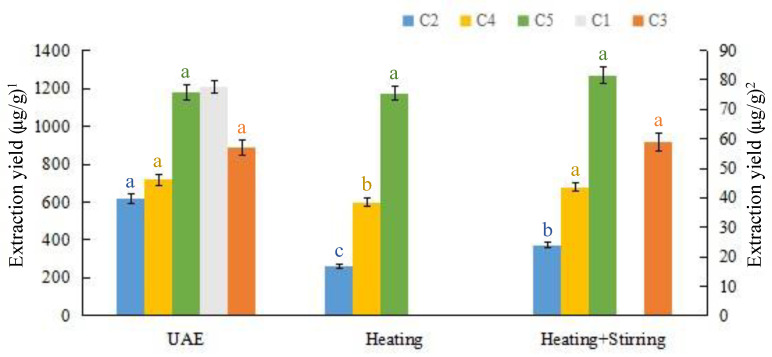
Comparison of the extraction efficiency between UAE and other extraction methods using DES-7. Different lowercase in columns of the same color represents a significant difference at *p* < 0.05. (^1^ Extraction yields of C2, C4 and C5; ^2^ Extraction yields of C1 and C3).

**Table 1 foods-12-00248-t001:** List of DESs synthesized and tested for macamide extraction.

Abbreviation	HBA	HBD	Molar Ratio
DES-1	Choline chloride	Levulinic acid	1:2
DES-2		Propanedioic acid	1:2
DES-3		1,4-Butanediol	1:2
DES-4		Glycerol	1:2
DES-5		Urea	1:2
DES-6		Triethylene glycol	1:2
DES-7		1,6-Hexanediol	1:2
DES-8		Xylitol	1:2
DES-9		DL-Malic acid	1:2
DES-10		Ethyl glycol	1:2

**Table 2 foods-12-00248-t002:** Box-Behnken design for the independent variables and the observed responses.

Run	*X*_1_ (mL/g)	*X*_2_ (°C)	*X*_3_ (min)	Extraction Yield (μg/g)				
				C1	C2	C3	C4	C5
1	0 (10:1)	0 (40)	0 (30)	77.50	602.14	57.08	719.40	1201.25
2	0 (10:1)	0 (40)	0 (30)	78.60	622.04	58.13	710.43	1197.92
3	−1 (8:1)	−1 (30)	0 (30)	71.14	504.41	49.35	568.32	1063.07
4	0 (10:1)	−1 (30)	1 (40)	67.69	486.67	43.71	607.95	962.07
5	0 (10:1)	1 (50)	1 (40)	68.32	491.35	46.65	584.90	1047.88
6	−1 (8:1)	0 (40)	−1 (20)	73.31	578.83	53.12	644.78	1123.30
7	0 (10:1)	0 (40)	0 (30)	80.40	606.05	56.97	705.52	1223.63
8	0 (10:1)	1 (50)	−1 (20)	66.69	486.54	44.66	580.04	936.22
9	0 (10:1)	0 (40)	0 (30)	79.02	598.54	54.50	707.97	1236.01
10	0 (10:1)	−1 (30)	−1 (20)	73.09	550.41	50.62	647.35	1142.50
11	0 (10:1)	0 (40)	0 (30)	80.82	613.87	56.69	699.71	1205.63
12	−1 (8:1)	1 (50)	0 (30)	65.71	463.57	42.64	528.33	953.95
13	−1 (8:1)	0 (40)	1 (40)	70.13	504.98	48.51	585.49	1066.53
14	1 (12:1)	0 (40)	1 (40)	68.98	490.85	45.46	543.04	964.98
15	1 (12:1)	0 (40)	−1 (20)	65.49	420.89	43.22	538.13	944.76
16	1 (12:1)	1 (50)	0 (30)	65.91	409.79	41.04	490.77	895.90
17	1 (12:1)	−1 (30)	0 (30)	64.71	412.15	41.83	502.41	916.16

**Table 3 foods-12-00248-t003:** ANOVA of the response surface quadratic model analysis for the extraction yield.

Variables	C1			C2			C3			C4			C5		
	Mean Square	*F*-Value	*p*-Value ^a^	Mean Square	*F*-Value	*p*-Value ^a^	Mean Square	*F*-Value	*p*-Value ^a^	Mean Square	*F*-Value	*p*-Value ^a^	Mean Square	*F*-Value	*p*-Value ^a^
Model	56.34	37.07	<0.0001	9358.95	76.79	<0.0001	62.52	35.71	<0.0001	10,744.64	106.55	<0.0001	25,878.04	94.97	<0.0001
*X* _1_	28.88	19.00	0.0033	12,649.25	103.79	<0.0001	60.89	34.78	0.0006	7973.95	79.08	<0.0001	29,409.19	107.93	<0.0001
*X* _2_	12.50	8.22	0.0241	1310.46	10.75	0.0135	13.83	7.90	0.0261	2520.15	24.99	0.0016	7803.13	28.64	0.0011
*X* _3_	1.50	0.98	0.3541	493.29	4.05	0.0841	6.64	3.79	0.0924	988.35	9.80	0.0166	1386.54	5.09	0.0387
*X* _1_ *X* _2_	10.99	7.23	0.0311	370.18	3.04	0.1249	8.76	5.00	0.0603	200.93	1.99	0.2009	1974.02	7.24	0.0310
*X* _1_ *X* _3_	11.12	7.32	0.0304	5170.33	42.42	0.0003	11.73	6.70	0.0360	1030.41	10.22	0.0151	1481.87	5.44	0.0325
*X* _2_ *X* _3_	12.36	8.13	0.0246	1174.78	9.64	0.0172	19.80	11.31	0.0120	489.74	4.86	0.0634	21,329.14	78.27	<0.0001
*X* _1_ ^2^	148.33	97.59	<0.0001	28,972.43	237.72	<0.0001	146.36	83.60	<0.0001	47,913.24	475.15	<0.0001	67,320.92	247.06	<0.0001
*X* _2_ ^2^	176.00	115.79	<0.0001	25,680.27	210.70	<0.0001	210.06	119.99	<0.0001	26,594.34	263.73	<0.0001	70,253.80	257.82	<0.0001
*X* _3_ ^2^	62.58	41.17	0.0004	2999.17	24.61	0.0016	43.14	24.64	0.0016	2439.79	24.19	0.0017	15,950.65	58.54	0.0001
Lack of fit	1.1100	0.6000	0.6464	165.1100	1.8500	0.2793	1.7200	0.9700	0.4900	165.6600	3.1700	0.1470	281.7500	1.0600	0.4586
*R* ^2^	0.9794			0.9900			0.9787			0.9928			0.9919		
Adj *R*^2^	0.9530			0.9771			0.9513			0.9834			0.9814		

^a ^*p*-values lower than 0.05 are statistically significant.

**Table 4 foods-12-00248-t004:** Comparison of extraction yields (μg/g) between petroleum ether and DES-7 for the five macamides.

Solvent	Extraction Yields (μg/g)				Total (μg/g)	
	C1	C2	C3	C4	C5	
Petroleum ether	60.93 ± 0.70 ^a^	420.73 ± 0.45 ^a^	48.75 ± 1.43 ^a^	593.99 ± 3.36 ^a^	776.32 ± 12.46 ^a^	1900.72 ± 9.64 ^a^
DES-7	79.27 ± 1.21 ^b^	608.53 ± 2.47 ^b^	56.67 ± 1.54 ^b^	708.61 ± 4.85 ^b^	1212.89 ± 15.96 ^b^	2665.97 ± 11.26 ^b^
Increment (%)	30.1	44.6	16.2	19.3	56.2	40.3

Note: Different lowercase in the same column represents a significant difference at *p* < 0.05.

**Table 5 foods-12-00248-t005:** Recovery yields of the five target macamides from the DESs.

Macroporous Resins	Yields (%)				
	C1	C2	C3	C4	C5
AB-8	82.75	79.26	75.40	85.32	87.88
DM-130	86.12	80.74	78.87	65.08	81.23
D-101	78.38	81.86	72.41	76.33	86.04
NKA	– ^a^	–	–	9.72	6.58
HPD-100	90.31	87.69	85.62	90.48	92.25
HP-20	–	–	–	–	69.41
D4020	77.56	–	79.68	–	–
DM-301	78.46	81.26	80.52	85.71	87.36

^a^ Below the limits of quantification, the recovery yields cannot be calculated.

**Table 6 foods-12-00248-t006:** The neuroprotective activities of the extracted macamide products on PC12 cells.

Group	Cell Viability (%)	Compared Group	*p*	Compared Group	*p*
Normal	100 ± 0.10	-	-	-	-
Model	52.14 ± 1.44	Normal	**	-	-
Nimodipine	87.98 ± 2.68	Model	***	-	-
DES–UAE (5 μg/mL)	76.51 ± 2.10	Model	***	-	-
PE-UAE (5 μg/mL)	68.38 ± 3.04	Model	**	DES–UAE (5 μg/mL)	*
DES-Heating (5 μg/mL)	58.16 ± 1.66	Model	**	DES–UAE (5 μg/mL)	***
DES-Heating + Stirring (5 μg/mL)	62.16 ± 1.32	Model	***	DES–UAE (5 μg/mL)	**
DES–UAE (10 μg/mL)	82.84 ± 1.68	Model	****	-	-
PE-UAE (10 μg/mL)	72.66 ± 1.87	Model	***	DES–UAE (10 μg/mL)	**
DES-Heating (10 μg/mL)	62.44 ± 1.72	Model	**	DES–UAE (10 μg/mL)	***
DES-Heating + Stirring (10 μg/mL)	68.75 ± 2.04	Model	***	DES–UAE (10 μg/mL)	***
DES–UAE (20 μg/mL)	90.67 ± 2.16	Model	****	-	-
PE-UAE (20 μg/mL)	86.12 ± 1.96	Model	****	DES–UAE (20 μg/mL)	ns
DES-Heating (20 μg/mL)	65.72 ± 1.88	Model	***	DES–UAE (20 μg/mL)	***
DES-Heating + Stirring (20 μg/mL)	74.89 ± 2.26	Model	***	DES–UAE (20 μg/mL)	***

Values are presented as mean ± S.D (*n* = 6). * *p* < 0.05, ** *p* < 0.01, *** *p* < 0.001, **** *p* < 0.0001.

**Table 7 foods-12-00248-t007:** The NO inhibition of the extracted macamide products in LPS-stimulated RAW264.7 cells.

Group	NO Inhibition (%)	Compared Group	*p*	Compared Group	*p*
L-Name	76.35 ± 0.86	-	-	-	-
DES–UAE (5 μg/mL)	50.46 ± 0.78	L-Name	****	-	-
PE-UAE (5 μg/mL)	38.16 ± 1.06	L-Name	****	DES–UAE (5 μg/mL)	***
DES-Heating (5 μg/mL)	18.65 ± 1.62	L-Name	****	DES–UAE (5 μg/mL)	***
DES-Heating + Stirring (5 μg/mL)	28.66 ± 1.25	L-Name	****	DES–UAE (5 μg/mL)	****
DES–UAE (10 μg/mL)	52.68 ± 1.39	L-Name	***	-	-
PE-UAE (10 μg/mL)	41.55 ± 1.46	L-Name	***	DES–UAE (10 μg/mL)	***
DES-Heating (10 μg/mL)	19.86 ± 1.53	L-Name	****	DES–UAE (10 μg/mL)	****
DES-Heating + Stirring (10 μg/mL)	30.58 ± 0.96	L-Name	****	DES–UAE (10 μg/mL)	****
DES–UAE (20 μg/mL)	55.73 ± 1.66	L-Name	**	-	-
PE-UAE (20 μg/mL)	43.81 ± 1.96	L-Name	***	DES–UAE (20 μg/mL)	**
DES-Heating (20 μg/mL)	21.33 ± 1.71	L-Name	****	DES–UAE (20 μg/mL)	****
DES-Heating + Stirring (20 μg/mL)	32.06 ± 1.64	L-Name	***	DES–UAE (20 μg/mL)	****

Values are presented as mean ± S.D (*n* = 6). ** *p* < 0.01, *** *p* < 0.001, **** *p* < 0.0001.

## Data Availability

Data are available within the article.

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
