# Peer review of "Deep Eutectic Solvent-Based Ultrasound-Assisted Strategy for Simultaneous Extraction of Five Macamides from Lepidium meyenii Walp and In Vitro Bioactivities"

_foods, 2023, doi:10.3390/foods12020248_

Round 1
Reviewer 1 Report
Comments and Suggestions for Authors
The presented work has an interesting objective but it is not focused on the field, its possible food use should be more highlighted in order to be published in this journal.
On the other hand, the structure of the paper does not correspond to the structure of a paper in which the important results are highlighted using tables and graphs that support it, but rather it looks more like an academic document in which the important parts of the study are not emphasised and the methodology and the results obtained are not correctly summarised.
In particular, the introduction section is very scattered without going into the importance of the compounds in the food. More emphasis is placed on the pharmacological importance and extraction techniques of the compounds. Figure 1 does not provide interesting information in the introduction.
In the material and chemicals section the paragraph from line 106 to 123 should be deleted. This is not usually included in scientific articles but the origin of the chemicals is mentioned in the methodologies section. Again this looks like an academic document and not a scientific one.
The extraction efficiencies obtained by single-factor optimisation and the Box-Behnken design are interesting, but the following section 2.6. does not explain a comparative study, but two methods to be used and then compared by some statistical treatment that is not clearly explained.
The section on results and discussion is very confusing and does not extract the relevant information. In general, the tables in the graphs sometimes present results that reveal little or nothing and are not selected to highlight what is relevant. Figure 2 should be presented in a table with the statistical treatment and correlations. Figure 3 is not necessary, the important results should be highlighted as all the graphs are practically the same. Figure 5 is difficult to understand and analysing the data is better seen in tables with the statistics. On the other hand, the discussion is very descriptive, and again not focused on relevance and application, nor on food. The activities tested discuss only the pharmacological relevance for the application in medicine.
The supplementary material is not relevant and in some cases like table S1 can be included in the tables of the paper.
Author Response
Reviewer 1
The presented work has an interesting objective but it is not focused on the field, its possible food use should be more highlighted in order to be published in this journal.
Response: Thanks to Reviewer for your comments. We have added the contents of maca extracts as possible food use in the Introduction, please see lines 54-59.
On the other hand, the structure of the paper does not correspond to the structure of a paper in which the important results are highlighted using tables and graphs that support it, but rather it looks more like an academic document in which the important parts of the study are not emphasised and the methodology and the results obtained are not correctly summarised.
Response: Thanks to Reviewer for your comments. We have revised the the structure of the paper.
In particular, the introduction section is very scattered without going into the importance of the compounds in the food. More emphasis is placed on the pharmacological importance and extraction techniques of the compounds.
Response: Thanks to Reviewer for your comments. We have added the contents of the possible use of macamides compounds as a food additive, please see lines 54-59.
Figure 1 does not provide interesting information in the introduction.
Response: Thanks to Reviewer for your comments. We have deleted Figure 1.
In the material and chemicals section the paragraph from line 106 to 123 should be deleted. This is not usually included in scientific articles but the origin of the chemicals is mentioned in the methodologies section. Again this looks like an academic document and not a scientific one.
Response: Thanks to Reviewer for your comments. We have revised and deleted the related section.
The extraction efficiencies obtained by single-factor optimization and the Box-Behnken design are interesting, but the following section 2.6. does not explain a comparative study, but two methods to be used and then compared by some statistical treatment that is not clearly explained.
Response: Thanks to Reviewer for your comments. In this study, we developed an effective method to extract macamides by DES-UAE. To highlight the extraction efficiency, we selected the other two common methods for comparison, and the data in Table 4 and Figure 3 have been added the statistical analysis.
The section on results and discussion is very confusing and does not extract the relevant information. In general, the tables in the graphs sometimes present results that reveal little or nothing and are not selected to highlight what is relevant.
Response: Thanks to Reviewer for your comments. In the results and discussion, we firstly discussed and demonstrated that the newly developed extraction method was an efficient way to prepare macamides extracts, the observed and predicted extraction yield of each macamide approved the process through the model verification. Then using the macamides extracts by DES-UAE, it showed superior neuroprotective and anti-inflammatory activities in in vitro assay. These finding revealed that the developed approach was a green and potential alternative that can be used to extract bioactive macamides constituents from L. meyenii in the food industries. So, the section of results and discussion is consistent.
Figure 2 should be presented in a table with the statistical treatment and correlations.
Response: Thanks to Reviewer for your comments. We have marked statistically significant differences using lowercase letters in Figure 2 (it is Figure 1 now).
Figure 3 is not necessary, the important results should be highlighted as all the graphs are practically the same.
Response: Thanks to Reviewer for your comments. Figure 3 (it is Figure 2 now) is the 3D response surface plots of the models for five macamides, we can observe the influence of each factor on the extraction yield of macamides distinctly, it is necessary, many similar articles put response surface plots in the text (Food Funct., 2019, 10, 1352; Food Chem., 2020, 319, 126555; Int. J. Biol. Macromol., 2017, 95, 675-681; Ind. Crop. Prod., 2020, 157, 112900).
Figure 5 is difficult to understand and analysing the data is better seen in tables with the statistics.
Response: Thanks to Reviewer for your comments. Figure 5 (it is Figure 4 now) showed the PC12 cells viability and NO inhibition in RAW264.7 cells of different macamides extracts, the differences have been marked in the figure, and we can observe the differences distinctly, it is better shown as figure than table, and is also a common representation of results for the similar assay in the literature.
On the other hand, the discussion is very descriptive, and again not focused on relevance and application, nor on food. The activities tested discuss only the pharmacological relevance for the application in medicine.
Response: Thanks to Reviewer for your comments. We have added the related contents for the possible food use in discussion, please see lines 430-445.
The supplementary material is not relevant and in some cases like table S1 can be included in the tables of the paper.
Response: Thanks to Reviewer for your comments. Table S1 showed the DESs that were not successfully prepared, it can give the readers some help for DES research and is more appropriate to put it in supplementary material. Table S3 showed the results of model verification. The figures in the supplementary material showed the results of some single-factor experimentations and comparison of the extraction efficiency. These are useful additions to the experimental procedure and results.

Reviewer 2 Report
Comments and Suggestions for Authors
This study investigates the effect of deep eutectic solvent-based ultrasound-assisted treatment on the extractibility of five Macamides from Lepidium meyenii Walp and their in vitro neuroprotective and anti-inflammatory activities. The topic of this study is very interesting and follows current trends in food technology - concerning the development of „green technologies” and their application in the food and pharmaceutical industries. The introduction, in general, provides a good background of the study and includes relevant references; however, provided the aim of the study requires additional information about performed in vitro activity assays. The experiments are well designed and described in sufficient detail, but the statistical analysis in the materials and methods wasn’t implemented (as subsection). The modes of results presentation are clear, however for some results is a lack of statistical analysis. The discussion is supported by results. In the discussion or conclusion section, the potential application of the obtained results, for the food industry, should be more emphasized (description of more specific usage) Conclusions summarizing the most important findings. In my opinion, this manuscript has quite good quality, but as described above needs some necessary improvements, especially concerning statistical analysis. Therefore I recommended a minor revision.
Detailed revisions:
Abstract
Line 23 – „The extracts of five macamides
showed superior pharmacological activities in vitro.” Please add more specific results.
Introduction
Line 85-95 – add information about in vitro activity assay in the aim of the study.
Materials and Methods
Statistical analysis should be described in Materials and Methods and performed for all assays. Add subsection „statistical analysis” at the end of this section.
Results
Line – 236 - „Results and discussion”?
Figure 2, 4 – mark statistically significant differences between means (represented by bars) e.g. using lowercase letters
Table 4 - mark statistically significant differences between means e.g. using superscript lowercase letters
Author Response
Reviewer 2
This study investigates the effect of deep eutectic solvent-based ultrasound-assisted treatment on the extractibility of five Macamides from Lepidium meyenii Walp and their in vitro neuroprotective and anti-inflammatory activities. The topic of this study is very interesting and follows current trends in food technology - concerning the development of “green technologies” and their application in the food and pharmaceutical industries. The introduction, in general, provides a good background of the study and includes relevant references; however, provided the aim of the study requires additional information about performed in vitro activity assays.
Response: Thanks to Reviewer for your comments. We have added the aim of in vitro activity assay of this study. Plerase see lines 95-101.
The experiments are well designed and described in sufficient detail, but the statistical analysis in the materials and methods wasn’t implemented (as subsection).
Response: Thanks to Reviewer for your comments. We have added the subsection “Statistical analysis” at the end of Materials and Methods.
The modes of results presentation are clear, however for some results is a lack of statistical analysis.
Response: Thanks to Reviewer for your comments. We have added statistical analysis in Figure 2, 4 and Table 4.
The discussion is supported by results. In the discussion or conclusion section, the potential application of the obtained results, for the food industry, should be more emphasized (description of more specific usage) Conclusions summarizing the most important findings. In my opinion, this manuscript has quite good quality, but as described above needs some necessary improvements, especially concerning statistical analysis. Therefore I recommended a minor revision.
Response: Thanks to Reviewer for your comments. We have added the potential application of the obtained results at the end of the Section 3.7, please see lines 430-445.
Detailed revisions:
Abstract
Line 23 – “The extracts of five macamides showed superior pharmacological activities in vitro.” Please add more specific results.
Response: Thanks to Reviewer for your comments. We have added the specific results, please see lines 23-26.
Introduction
Line 85-95 – add information about in vitro activity assay in the aim of the study.
Response: Thanks to Reviewer for your comments. We have added the aim of in vitro activity assay of this study. Plerase see lines 95-101.
Materials and Methods
Statistical analysis should be described in Materials and Methods and performed for all assays. Add subsection “statistical analysis” at the end of this section.
Response: Thanks to Reviewer for your comments. We have added the subsection “Statistical analysis” at the end of Materials and Methods.
Results
Line – 236 – “Results and discussion”?
Response: Thanks to Reviewer for your comments. We have corrected to “Results and discussion”.
Figure 2, 4 – mark statistically significant differences between means (represented by bars) e.g. using lowercase letters
Response: Thanks to Reviewer for your comments. We have marked statistically significant differences using lowercase letters.
Table 4 - mark statistically significant differences between means e.g. using superscript lowercase letters
Response: Thanks to Reviewer for your comments. We have marked statistically significant differences in Table 4 using superscript lowercase letters.

Reviewer 3 Report
Comments and Suggestions for Authors
The manuscript titled Deep Eutectic Solvent-based Ultrasound-assisted Strategy for Simultaneous Extraction of Five Macamides from Lepidium meyenii Walp and in vitro Bioactivities is interesting work and in my opinion it should get major revision.
1) Validation of HPLC method is missing.
2)Table 4 is not clear. Perhaps the units could be change.
3) The aim of the manuscript was to develope new method of extraction. Why did you decide to determine biological activity in this study? How can you connect biological activity with new method extraction?
4) Why did you choose "maca"?
Author Response
Reviewer 3
The manuscript titled Deep Eutectic Solvent-based Ultrasound-assisted Strategy for Simultaneous Extraction of Five Macamides from Lepidium meyenii Walp and in vitro Bioactivities is interesting work and in my opinion it should get major revision.
Comment No. 1: Validation of HPLC method is missing.
Response: Thanks to Reviewer for reminder. In the Section 2.3, we listed the calibration linearity, precision and accuracy of the HPLC method, the other items were not shown. We have added all the HPLC method validation, the results were shown in Section 3.3 and the Table S2 in Supplementary Material.
Comment No. 2: Table 4 is not clear. Perhaps the units could be change.
Response: Thanks to Reviewer for your question. The contents of the compounds C1 and C3 were lower than 100 μg/g, for the accuracy of the data and the overall consistency with the other 3 compounds, the results were presented as the units of μg/g.
Comment No. 3: The aim of the manuscript was to develop new method of extraction. Why did you decide to determine biological activity in this study? How can you connect biological activity with new method extraction?
Response: Thanks to Reviewer for your question. In this study, we developed a new method to extract five major macamides from maca roots firstly, and then the extracted macamide products of DES-UAE were compared with other extraction procedure such as PE-UAE, DES-UAE, DES-Heating and DES-Heating + Stirring. According to the results of biological activity, macamide products of DES-UAE possessed higher neuroprotection and anti-inflammatory activities than others, this revealed that we not only developed a new extraction method for macamides extraction, but also the obtained extracts by the method had stronger activities. This showed thoroughly that our newly established method had greater application value. So, there is a correlation between extraction process and activity.
Comment No. 4: Why did you choose "maca"?
Response: Thanks to Reviewer for your question. As we have stated in the text, maca root powder has been formally approved to be included in the New Resource Food Directory by the Chinese Ministry of Health in 2011, it is an edible medicinal plant in Peruvian tradition and has been used as a food source for centuries. Maca contained bioactive compounds such as alkaloids, glucosinolates, fatty acids, phenols, and polysaccharides. Researchers have focused on the effects of using a powder or extract of maca as a food additive for humans and livestock diets. In addition, in populations living in the Peruvian Central Andes over 4000 meters of altitude, maca consumption has been associated with low serum Interleukin-6 levels and, in turn, with better health status and low chronic mountain sickness scores. Maca extracts will be increasingly widely used in the diets and pharmaceutical industries. Although the preparation methods of maca extracts have been reported, they mainly used organic solvents, there are limited studies on the use of green solvents such as deep eutectic solvent. We developed a new method based on deep eutectic solvent for the preparation of macamides extracts.

Round 2
Reviewer 1 Report
Comments and Suggestions for Authors
Some modifications have been introduced, however:
In section 2.6. the comparison is not explained, but two methods are explained and then compared. As already indicated in the previous revision, the title has to be changed to0: 22.6. Other extraction methods for comparison”
Figure 3 is not adequately highlighted. As I remarked in the previous review it would be important to highlight and detail the specific results of time-temperature and extraction yield, since all the graphs are very similar. Explain and detail more in the text.
Figure 5 is again hard to understand in terms of the differences shown. This figure, if the authors do not want to transform it into a table, should be more clearly explained. The statistics are, in my opinion, more clearly reflected in tables.
Some of the data in the supplementary material are still not useful and can be explained in the text of the article, e.g. Table S1 with the DES that were not successfully prepared.
Author Response
Some modifications have been introduced, however:
Comment No. 1: In section 2.6. the comparison is not explained, but two methods are explained and then compared. As already indicated in the previous revision, the title has to be changed to0: 22.6. Other extraction methods for comparison”
Response: Thanks to Reviewer for your valuable suggestions. We have changed the title to be “Other extraction methods for comparison” in section 2.6.
Comment No. 1: Figure 3 is not adequately highlighted. As I remarked in the previous review it would be important to highlight and detail the specific results of time-temperature and extraction yield, since all the graphs are very similar. Explain and detail more in the text.
Response: Thanks to Reviewer for your valuable suggestions. We have added more discussion as for the Figure of 3D surface plots, please see lines 325-350.
Comment No. 1: Figure 5 is again hard to understand in terms of the differences shown. This figure, if the authors do not want to transform it into a table, should be more clearly explained. The statistics are, in my opinion, more clearly reflected in tables.
Response: Thanks to Reviewer for your valuable suggestions. We have transformed Figure 5 into Table 6 and Table 7.
Comment No. 1: Some of the data in the supplementary material are still not useful and can be explained in the text of the article, e.g. Table S1 with the DES that were not successfully prepared.
Response: Thanks to Reviewer for your valuable suggestions. We have deleted Table S1 in the supplementary material.

Reviewer 3 Report
Comments and Suggestions for Authors
The authors corrected and improved their manuscript. I accept their answers.In my opinion the manuscript can be published in Foods.
Author Response
Comment: The authors corrected and improved their manuscript. I accept their answers. In my opinion the manuscript can be published in Foods.
Response: Thanks to Reviewer for your valuable feedback that we have used to improve the quality of our manuscript.